# Effects of Terpenes on the Osteoarthritis Cytokine Profile by Modulation of IL-6: Double Face versus Dark Knight?

**DOI:** 10.3390/biology12081061

**Published:** 2023-07-28

**Authors:** Giacomo Farì, Marisa Megna, Salvatore Scacco, Maurizio Ranieri, Maria Vittoria Raele, Enrica Chiaia Noya, Dario Macchiarola, Francesco Paolo Bianchi, Davide Carati, Antonio Gnoni, Alessio Danilo Inchingolo, Erda Qorri, Antonio Scarano, Antonio Scacco, Roberto Arrigoni, Biagio Rapone

**Affiliations:** 1Department of Translational Biomedicine and Neuroscience (DiBraiN), Aldo Moro University, 70121 Bari, Italy; marisa.megna@uniba.it (M.M.); salvatore.scacco@uniba.it (S.S.); maurizio.ranieri@uniba.it (M.R.); maryvi.92@hotmail.it (M.V.R.); enrica.chiaianoya@gmail.com (E.C.N.); dmacchiar@gmail.com (D.M.); antonio.gnoni@uniba.it (A.G.); 2Department of Biological and Environmental Science and Technologies (Di.S.Te.B.A.), University of Salento, Piazza Tancredi 7, 73100 Lecce, Italy; 3Mater Dei Hospital C.B.H., 70125 Bari, Italy; 4Istituti Clinici Scientifici Maugeri, IRCCS, 70124 Bari, Italy; scaccoantonio0@gmail.com; 5Department of Interdisciplinary Medicine, Aldo Moro University of Bari, 70121 Bari, Italy; dr.francesco.bianchi@gmail.com (F.P.B.); ad.inchingolo@libero.it (A.D.I.); biagiorapone79@gmail.com (B.R.); 6Ansce Bio Generic, 73020 Carpignano Salentino, Italy; davide.carati@anscebiogeneric.com; 7Dean Faculty of Medical Sciences, Albanian University, Bulevardi Zogu I, 1001 Tirana, Albania; e.qorri@albaninanuniversity.edu.al; 8Department of Innovative Technologies in Medicine and Dentistry, University of Chieti-Pescara, 66100 Chieti, Italy; ascarano@unich.it; 9CNR Institute of Biomembranes, Bioenergetics and Molecular Biotechnologies (IBIOM), 70125 Bari, Italy; r.arrigoni@ibiom.cnr.it

**Keywords:** IL-6, myokines, osteoarthritis of the knee, dietary supplement, nutraceuticals, hemp seed oil, terpenes

## Abstract

**Simple Summary:**

Terpenes are emerging as a new complementary therapy for several chronic diseases, but their effects on inflammatory cytokines are not fully clear. The aim of this study was to deepen the understanding of how a hemp seed oil and terpene-based dietary supplement modifies blood cytokine levels in a population of patients suffering from knee osteoarthritis, in comparison with a similar one containing hemp seed oil only. A total of 38 patients were selected using specific criteria and subsequently divided into two subgroups of 19 patients each that underwent a 45-day treatment with these two different dietary supplements. Blood samples were taken immediately before and after this therapy. The group which received hemp seed oil and terpenes had a significant rise in their IL-6 levels and a significant decrease in their IL-1β levels. These blood changes could be the biochemical basis on which to build an anti-inflammatory action at the joint level, consequently opening up new scenarios to follow in the integrative therapies of osteoarthritis.

**Abstract:**

Background: Hemp seed oil and terpenes are emerging as a dietary supplement and complementary therapy for patients suffering from knee osteoarthritis (KOA). However, the mechanisms and effects induced by these molecules on inflammatory cytokines are not yet fully understood. The aim of this study was to evaluate the changes in the cytokine IL-1β, IL-1α, IL-2, IL-6, and TNF-α levels from two oral hemp seed oil-based dietary supplements, of which only one included the addition of terpenes, in a population of KOA patients. Methods: Sera from venous blood samples were collected from thirty-eight patients who were divided into two subgroups. The control group underwent a 45-day treatment with a dietary supplement containing only hemp seed oil, while the treatment group assumed a hemp seed oil and terpene-based dietary supplement for the same number of days. A Bio-Plex Human Cytokine assay was performed by a customized human cytokine five-plex panel for IL-1β, IL-1α, IL-2, IL-6, and TNF-α. Patients were evaluated before the beginning of the treatment (T0) and soon after it (T1). Results: No measurable levels of IL-2 and TNF-α were found in any of the subjects. Low levels of IL-1β were found, which were significantly decreased in the treatment group. No change in IL-1α levels was observed, while treated patients had a significant increase in IL-6 levels. Conclusions: Hemp seed oil and terpene treatment modified the IL-1β and IL-6 levels, counteracting KOA inflammation in this way. In this study, IL-6 revealed its new and alternative action, since it is traditionally known as a pro-inflammatory factor, but it recently has been found to have anti-inflammatory activity in the muscle-derived form, which is the one it assumes as a myokine when activated by terpenes.

## 1. Introduction

Knee osteoarthritis (KOA) is one of the most common degenerative diseases causing disability in elderly patients [1] and, in recent years, this condition has been increasing in younger subjects aged 45 to 70 years [2]. It is a chronic joint disease [3] caused by the slow degradation of articular cartilage that, over time, leads to severe forms of joint deformity and pain, with consequent functional impairment [4]. KOA remains, nowadays, the greatest challenge in the field of osteoarthritis, with a high morbidity load and unavailable definitive solutions in terms of treatments [5], heavily affecting health care costs all around the world, especially in the most industrialized countries’ populations [6]. Early symptomatic cases should be promptly diagnosed in order to intervene with proper management, including structured education, therapeutic exercise, weight management, and dealing with lifestyle-related risk factors to slow the progression of the disease [7]. Many factors have been involved in the development of KOA, including cytokines and soluble mediators, and it is feasible that therapies able to modulate these factors could be effective in the treatment of KOA [8,9]. This is the reason why it is necessary to focus on certain molecules that can appropriately manage and modify inflammatory cytokine levels in the blood of some affected patients.

Recently, the use of nutraceuticals has been gaining popularity in health condition improvement, preventing chronic diseases, delaying aging, and supporting many body functions [10]. Several studies have already shown the potential role of nutraceuticals in the treatment of pathological conditions in bone and soft tissues, such as articular cartilage. These studies have now proven that polyunsaturated fatty acids (n-3 PUFAs) are able to counteract the onset and the progression of osteoarthritis (OA) by reducing bone and cartilage destruction, inhibiting proinflammatory cytokines release, reactive oxygen species (ROS) generation, and the nuclear factor kappa-light-chain-enhancing of the activated B cells (NF-kB) pathway’s activation by reducing bone and cartilage destruction, inhibiting proinflammatory cytokines release, reactive oxygen species (ROS) generation, and the nuclear factor kappa-light-chain-enhancer of the activated B cells (NF-kB) pathway’s activation [11].

Since the Cannabis Sativa plant has been used for a long time for medical and recreational purposes and has been shown to be effective in several diseases such as pain, epilepsy, sickness, and vomiting and psychiatric conditions such as anxiety disorders and post-traumatic stress [12,13,14], scientists decided to learn more about its individual components. Among the 550 chemical compounds already found in cannabis, more than 100 phytocannabinoids were identified, including D9-tetrahydrocannabinol (D9-THC) and cannabidiol (CBD) [15] which have recently shown interesting pain-relieving properties in preclinical and clinical studies, both in animals and in humans [16,17]. Cannabis also contains aromatic terpenes, which are natural molecules of great interest for their therapeutic benefits in the management of chronic diseases. Terpenes are cyclic or alicyclic hydrocarbons whose structure consists of an even number of isoprene units bound together in a characteristic manner [18]; terpenes are widely found in essential oils and natural resins and are broadly used in the pharmaceutical and chemical industries [19]. In recent years, the international scientific community has been devoted to the in-depth research of each individual terpene in order to find their respective possible therapeutic applications and, to date, many of these substances already have countless uses in the management of several conditions. Among the most studied terpenes are α- and β-pinene, mainly used as antibiotic resistance modulators, anticoagulants, and antitumoral and antimicrobial agents [20]; limonene, which is known for its neuroprotective potential even in severe and widespread diseases such as multiple sclerosis, Alzheimer’s disease, and stroke [21]; α-terpinene, with its anti-parasitic effect against major pathogens, such as Plasmodium falciparum, Leishmania spp., Trypanosoma spp., and Trichomonas vaginalis, due to its anti-histamine and anti-acetylcholinesterase activity [22]; and sesquiterpene valencene [23] and particularly, β-caryophyllene (BCP) and β-Myrcene, for their in vivo and in silico anti-inflammatory properties [24]. BCP is a bicyclic sesquiterpene widely distributed in the plant kingdom [25] that demonstrated selective action on the CB2 endocannabinoid receptor (CB2 receptor) and attracted considerable attention because of its several pharmacological activities, low toxicity, and safety profile [26]. On the other hand, β-Myrcene is a natural phytochemical monoterpene that possesses potent anti-inflammatory activity; it is already widely used in the treatment of inflammatory bowel diseases (IBDs) [27] and for protecting the liver against acetaminophen-induced injury by reducing oxidative stress and the inflammatory response [28].

Both BCP and β-Myrcene represent a new therapeutic opportunity for pain treatment [19] and have been shown to be promising in improving knee joint function in patients affected by local osteoarthritis [29]. While their clinical beneficial effects have previously been shown, the mechanisms and specifically the biochemical actions of BCP and β-Myrcene oral administration still remain unclear. Therefore, the aim of this study is to analyze how inflammatory cytokine {interleukin-1α (IL-1α), interleukin-1β (IL-1β), interleukin-2 (IL-2), interleukin-6 (IL-6) and tumor necrosis factor-α (TNF-α)} serum levels change when two regimens of oral hemp seed oil-based dietary supplements, of which only one includes the addition of terpenes (BCP and β-Myrcene), are administered in a population of KOA patients.

## 2. Materials and Methods

Our study is a double-blind prospective cohort study, and it was conducted between March and August 2022.

Thirty-eight patients with monolateral KOA and who fulfilled specific enrollment criteria were enrolled. The patients were then divided into two groups, the control group and the treatment group, consisting of nineteen subjects each.

The inclusion criteria were: patients had to be aged between 45 and 70 years; they had to be affected by KOA according to the American College of Rheumatology criteria; only patients with a Numeric Rating Scale (NRS) ≥ 4 at the enrollment time and in the previous 15 days were included; they had to have a well-documented KOA grade II-III according to the Kellgren–Lawrence scale; only patients who were able to understand the aim of the study, the possible side effects, and were able to provide informed consent were enrolled. The exclusion criteria were the following: patients presenting any local complications at knee level (e.g., hematoma or swollen joint and knee deformation); knee pain due to trauma (on the examination day or during the previous three months); patients affected by any other medical pathology that could potentially interfere with the outcome of the study (e.g., rheumatological diseases such as rheumatoid arthritis, systemic lupus erythematosus, ankylosing spondylitis, and metabolic inflammatory arthropathy); patients who underwent local drug infiltration (including corticosteroids, prolotherapy, viscosupplementation, platelet-rich plasma, polynucleotides, and/or stem cells) or physiotherapy within the previous 45 days (including both therapeutic exercises and any physical therapy); patients who assumed any other treatment in the previous 15 days before the enrollment such as non-steroidal anti-inflammatory drugs (FANS), analgesics, or steroids were excluded from the study due to the risk of interfering with the outcome; patients who assumed any type of slow-acting drugs or dietary supplements in the previous three months before the enrollment were excluded; patients with contraindications to acetaminophen were not allowed in the study; systemic medical conditions which contraindicate nutraceutical assumption (e.g., kidney or liver failure, cardiovascular disease not well controlled, and/or inflammatory bowel diseases such as Crohn’s disease, ulcerative rectocolitis, and irritable bowel syndrome); moreover, pregnant or breastfeeding women were excluded; pre-menopausal women without contraception; and patients who participated in any other clinical trials within the previous three months before the enrollment.

At the time of enrollment (T0), the medical history and informed consent were collected. All patients were subjected to a standardized physical examination and knee X-ray evaluation. Then, their Body Mass Index (BMI) was calculated according to the formula: weight (Kg)/height (m^2^) after the detection of their weight and height. On the same day, all patients were subjected to a blood draw to dose the following serum inflammatory cytokines: IL-1α, IL-1β, IL-2, IL-6, and TNF-α. The serum sample cytokine assay was measured by using a customized Bio-Plex Pro Human Cytokine 5-plex assay (IL-1β, IL-1α, IL-2, IL-6, and TNF-α) (CAT#12015899, Biorad Laboratories, Hercules, CA, USA). The assay was carried out according to the manufacturer’s instructions and analyzed by the Bio-Plex 200 system and Bio-Plex Manager software v. 6.1.1 (Biorad Laboratories, USA).

All thirty-eight patients underwent 45 days of treatment.

Specifically, patients belonging to the control group underwent a 45-day treatment with a hemp seed oil-based dietary supplement (without cannabinoids), in a soft gel format. They took one capsule at lunch and one capsule at dinner (main meals) for a total of two soft gel capsules per day. On the other hand, patients belonging to the treatment group underwent a 45-day treatment with a hemp seed oil-based dietary supplement also containing terpenes, β-caryophyllene and β-myrcene specifically (still without cannabinoids). They also took one capsule at lunch and one capsule at dinner (main meals) for a total of two soft gel capsules per day. The hemp seed oil contained in both dietary supplements was composed mainly of linoleic (55.90%), gamma Linolenic (19.10%), and oleic (9.30%) acids.

The design of the study can be considered a double-blind study because both the patients and the investigators who evaluated them did not know which of the two dietary supplements had been administrated; this was made possible by the fact that both supplements had the same confection and were, therefore, not recognizable. For this reason, the distribution of the dietary supplements was handled by a third investigator.

Each patient was provided with a dedicated diary to record the intake of any other medical drugs during the 45-day treatment period. For pain management, paracetamol (up to a maximum of 3000 mg/day) was allowed. All the diaries were collected at the end of the treatment (T1).

At T1, 45 days after T0, all patients underwent a new blood draw in order to compare the blood chemistry profiles between the two groups.

No patients reported any side effects of the therapy.

All the necessary information was given to the patients during the first medical examination. On the same occasion, the written informed consents were collected. All the performed procedures were carried out in accordance with the Helsinki Declaration (2016) of the World Medical Association; the study was approved by the Ethics Committee of Albania University, Tiran, Albania (Nr. 587 Prot.—Date: 13 December 2021).

### Statistical Analysis

The absolute values of the serum cytokines were expressed as nanograms/dL, unless not available (n.a.). For each group, the mean values and standard mean errors were calculated. Finally, the percentage (%) variation of cytokine levels between the values collected at T1 and T0 (expressed as 100%) was calculated. The Student’s *t*-test was used to report the significance of variation (*p*) between the T0 and T1 groups, which were indicated as not significant (NS), *p* < 0.05 (*), *p* < 0.01 (**), and *p* < 0.001 (***).

## 3. Results

The study sample was made up of a total of 38 subjects, of which 19 patients (50.0%) belonged to the control group and 19 patients (50.0%) to the treatment one; the characteristics of the sample, by group, are shown in Table 1.

In all subjects, the Bio-Plex Pro Human Cytokine 5-plex assay (IL-1β, IL-1α, IL-2, IL-6, and TNF-α) did not find measurable levels of IL-2 and TNF-α. Only IL-1β, IL-1α, and IL-6 were detected by the Bio-Plex Pro Human Cytokine 5-plex assay. The IL-1β levels were low in all subjects and treated patients, a significant decrease was observed. IL-1α did not show relevant modifications. The levels of IL-6 in the control group were not modified, whilst the treatment group showed a significant increase in IL-6. All the above-mentioned results are described in Table 2 and Figure 1.

Only a random analgesic intake emerged from the analysis of the analgesic intake diaries, which settled on an average of 1.0 g/week per group, with a sporadic and not significant distribution among the participants. There was a complete absence of side effects reported.

## 4. Discussion

These findings show that patients belonging to the treatment group had a significant change and a global reduction in the expression of pro-inflammatory cytokines compared to the control group (Table 2, Figure 1).

It is traditionally known that OA pathophysiology is complex and influenced by many factors that converge in a single disease model of a purely inflammatory nature. The pathogenesis of OA is therefore based on the action of secreted inflammatory molecules, such as proinflammatory cytokines, which are the most important mediators of all the processes implicated in this chronic degenerative disease. In fact, pro-inflammatory cytokines, together with nitric oxide, prostaglandin E2, and neuropeptides, are responsible for articular cartilage matrix degeneration. They are produced by the inflamed synovial tissue and upset the physiological balance between repair and degradation, ensuring catabolic processes prevail thanks to the action of lytic enzymes which are able to destroy the cartilage [30]. Although, from a clinical point of view, OA presents phases of quiescence and phases of reactivation, the biochemical mechanisms that underlie it do nothing but smolder, giving this pathology a chronic degenerative evolution due to the ineluctable inflammation that modifies the physiological functioning of joint cells, favoring chondrocyte apoptosis and the activation of macrophages.

This inflammatory process then inevitably extends to the entire joint microenvironment, leading to the formation of geodes and osteophytes and a reduction in the joint space; swelling, deformity, and joint pain thus cause a functional limitation that can determine severe forms of motor disability and can negatively impact the physical and mental wellbeing of the patients, especially in its advanced stages, which are typically related to old age.

The above description can be valid for all joints, starting with the arthritic knee. The knee joint, indeed, is certainly the most affected by OA, followed by the hip, hands, and spine. For this reason, it is also traditionally the most investigated, given that it may represent the first site of the manifestation of OA itself, even in isolated forms [31]. Although the knee is a joint subjected to a continuous and intense load, and, consequently, it is more exposed to the risk of joint degeneration in the presence of predisposing factors such as obesity and reduced muscle tone and strength, it is now also well known that the OA-related biomechanical factors in this case go strictly hand-in-hand with the biochemical ones. In a recent review by P. Dainese et al., the association between inflammation (as measured by effusion, synovitis, baker’s cysts, cytokines, and the C-reactive protein) and pain in patients with radiographic KOA was investigated [32]. Despite varying results in terms of the strength of evidence, what emerges is that both imaging tests and blood tests confirmed the inflammatory nature of KOA, which therefore results in joint pain.

As a result, the modulation of the proinflammatory cytokine profile represents a possible new target for new OA treatments [33]. The real challenge is to find increasingly selective and innovative solutions which allow for personalized therapies, amplifying the benefits and reducing the side effects of the drugs currently used to fight inflammation. In fact, non-steroidal anti-inflammatory drugs are effective but burdened by side effects such as an increase in blood pressure, variations in blood sugar, irritation to the gastric mucosa, and negative changes in the coagulation profile. This imposes severe limitations on their use and calls for the scientific community to make an effort in the search for effective but well-tolerable molecules, both topically and systemically. As far as joint injections are concerned, hyaluronic acid remains a reliable solution with long-lasting effects over time; nevertheless, regenerative medicine is opening up to new solutions such as platelet-rich plasma and stem cells, with very encouraging clinical results, and even nanomedicine, an interdisciplinary discipline which aims to manipulate particles the size of 1–100 nanometers, is studying the possibility of using new bioparticles such as exosomes, dendrimers, micelles, and lipid vesicles as transports for drug release control [34]. In the near future, this could avoid excessive infiltration practices, which are minimally invasive procedures that are poorly tolerated by some patients, especially in the presence of coagulopathies and systemic diseases that expose them to the risk of infections due to a low white blood cell count. In this sense, experimenting with natural molecules that can be taken orally and that can develop an anti-inflammatory power without significant side effects is still an open game that researchers and clinical physicians are necessarily called upon to play.

This is the reason why this research aimed to deepen the understanding of the effects of dietary supplements containing hemp seed oil and terpenes in the inflammatory cytokine expression in a population of KOA patients.

With regard to the IL-2 and TNF-α levels, these were not measurable. This can be explained by the exclusion criteria which required the absence of systemic chronic inflammatory diseases. In fact, many cytokines of the IL-2 family as well as TNF-α have been reported to be a driving force in immune cell activation and systemic inflammation [35,36,37]. The enrollment criteria predicted that patients were suffering from monolateral KOA, which can be considered a district pathology, and that they were not affected by rheumatoid arthritis (RA), metabolic inflammatory arthropathy, or any other chronic inflammatory systemic diseases. In fact, these conditions are associated with a general activation of several pro-inflammatory cytokines. In a study conducted by Kondo N et al. in 2021, the role of TNF-α in the pathogenesis of RA was demonstrated and an increased expression of TNF-α was documented in affected patients, inducing autoimmune arthritis in transgenic animals [38]. Moreover, the increased serum level of these cytokines in chronic systemic inflammatory diseases such as vasculitis [39], psoriasis [40], and long COVID-19 syndrome [41] is now clear. This explains why, in our subjects, the Bio-Plex Pro Human Cytokine 5-plex assay (IL-1β, IL-1α, IL-2, IL-6, and TNF-α) did not find measurable levels of the pro-inflammatory cytokines IL-2 and TNF-α and low levels of IL-1β.

IL-1β levels decreased between T0 and T1 in both groups, but significantly more in the treatment group. IL-1 family members are the most highly profiled in OA and have all been shown to be present in the synovial fluid and subchondral bone of OA patients [42]. IL-1β is also actively involved in the pathogenesis of cartilage loss related to OA [43]. The low levels of IL-1β induced by the local inflammatory process in KOA [44] were further reduced by hemp seed oil and terpenes, which therefore demonstrated their synergic anti-inflammatory action. Terpenes’ healing potential for OA is rapidly being documented. In a 2021 study, the therapeutic effect of limonin on OA was assessed in chondrocytes in vitro in IL-1β-induced OA and in the destabilization of the medial meniscus (DMM) mice in vivo [45]. Similarly, aucubin, a natural compound isolated from Eucommia ulmoides, inhibits IL-1β-induced chondrocyte apoptosis [46] and morusin ameliorates IL-1β-induced chondrocyte inflammation and OA via the NF-κB signal pathway [47].

Although further studies are needed to confirm the clinical effectiveness of different terpenes, there is no doubt that these molecules can now be considered as a complementary therapy to OA by interfering with IL-1β.

Although a minor increase in its levels could be glimpsed in the treatment group between T0 and T1, the IL-1α levels did not have any statistically significant modifications between the two groups at the detection times. Also, these data are in line with the available literature, since the most recent evidence suggests that the role of IL-1α in OA pathogenesis is still controversial [48]. However, it seems that an IL-1α increase could have a protective action for OA patients, though at the moment it is not possible to unequivocally establish the role of this cytokine in joint inflammatory processes [49].

The particular case of IL-6 caught our attention. Indeed, this interleukin decreased slightly between T0 and T1 in the control group, while it significantly increased in patients in the treatment group after 45 days of treatment.

This suggests an unexpected action of the IL-6 cytokine, generally known as a pro-inflammatory factor, which was recently found to have anti-inflammatory activity in its muscle-derived form, which could be induced by hemp seed oil and in particular by terpenes (Figure 1).

Interestingly, hemp seed oil and terpene treatment increased the levels of IL-6, which is a cytokine with established pro- and anti-inflammatory action [50]. In classic signaling, IL-6 induces intracellular signaling pathways after binding to its membrane-bound receptor (IL-6R) expressed on hepatocytes and certain subpopulations of leukocytes mediated by the membrane-bound b-receptor glycoprotein 130 (gp130) [51]. In a second pathway, which is called trans-signaling, IL-6 binds to soluble forms of the IL-6R (sIL-6R), and this agonistic IL-6/sIL-6R complex can, in principle, activate all cells due to the uniform expression of gp130 [52]. Most pro-inflammatory roles of IL-6 have been attributed to the trans-signaling pathway, whereas anti-inflammatory and regenerative signaling is mediated by IL-6 classic signaling.

This double face characteristic of IL-6 can explain its anti/pro-inflammatory activity in different conditions.

Systemic chronic inflammation involves progressive changes in inflammatory cells and the coexistence of tissue destruction and repair; it can become pathological because of the loss of tolerance or because of regulatory processes. Obesity and metabolic syndrome lead to an inflammatory state that differs from the classical response, with the inflammatory process being systemic and characterized by a chronic low-intensity reaction [53,54]. Chronic systemic inflammatory diseases like obesity, non-alcoholic fatty liver disease/nonalcoholic steatohepatitis (NAFLD/NASH) liver diseases, diabetes, cardiovascular diseases, pain, and other nervous system disorders are related to a mild chronic inflammation, which appears to start from an alteration in the metabolism of these patients and, for this reason, it is also named metaflammation, that occurs in several tissues, including adipose, pancreas, liver, muscle, brain, and heart tissues [55]. In these chronic inflammatory diseases, the use of bioactive compounds acting on CB2 cannabinoid receptors has shown therapeutic effects through the reduction in pro-inflammatory mediators such as TNF-α, IL-1β, and IL-6 [56,57] (Figure 2).

KOA is a chronic joint disease whose mechanisms of progression are not fully understood [4], but it is evidently the expression of a local disease rather than a chronic systemic one. As said above, this is also evidenced by the fact that, in the pre-surgical stages, treatment involves the on-site injection of substances such as corticosteroids [58], hyaluronic acid [59], platelet-rich plasma [60], or oxygen–ozone [61]. It has recently emerged that the role of muscle-derived IL-6 is unlike what was assumed in the past. Interestingly, it has been indicated that IL-6 plays opposing roles as an anti-inflammatory myokine and as a pro-inflammatory cytokine [25,62]. Skeletal muscles can release several bioactive substances, collectively named myokines, among which IL-6 is the most abundant. The endocannabinoid system (ECS) is involved in various processes, including brain plasticity, learning and memory, neuronal development, nociception, inflammation, appetite regulation, digestion, metabolism, energy balance, motility, and the regulation of stress and emotions [63]. The activation of CB2 cannabinoid receptors appears to induce the release of several mediators from muscle tissue which have been defined as exerkines, including IL-6 [64], that have anti-inflammatory activity. As a consequence, this change in IL-6 serum levels enforces the hypothesis that hemp seed oil and terpenes could represent a valid option in the complementary therapy of KOA. This could, therefore, become a new opportunity to personalize therapies, making the individual rehabilitation project more effective and multidimensional [64].

However, this study cannot be considered free of limitations. The main limitation is the short duration of the follow-up and the small patient sample. Therefore, additional studies are needed to monitor how serum cytokine levels change over time and how this finding affects pain and joint function. In addition, it should be considered that inflammatory cytokine serum levels depend on several factors such as smoking, diet, sedentary lifestyle, or frequent sport activity. However, it should be pointed out that in this study we were certain that these parameters remained constant in the sample analyzed between T0 and T1. In addition, we did not investigate the correlation between the blood chemistry data and the clinical data of these patients, but we believed it was essential in the first instance to verify the blood variations induced by terpenes; however, we intend to follow up on this research with further studies that also will include appropriate clinical evaluations. Finally, there was no placebo group, but this choice was due to the necessity of guaranteeing treatment for all patients since all of them suffered from significant knee pain. Nevertheless, future studies will overcome this limitation, in compliance with the necessary ethical rules.

On the contrary, we consider an important strength of this study is the fact that it highlighted the new and probably unexpected anti-inflammatory effects of terpenes, with particular reference to IL-6, thus opening further research scenarios on the therapeutic potential of these molecules.

## 5. Conclusions

In conclusion, terpenes (β-caryophyllene and myrcene) and hemp seed oil (without cannabinoids) could counteract KOA inflammation by reason of a synergic action of IL-6 induction and IL-1α reduction. Further studies are needed to understand the duration of these effects over time and to translate them into significant changes in the clinical picture of the patients, but these interesting findings could represent the starting point for new studies about the in vivo anti-inflammatory action of these promising dietary supplements.

## Figures and Tables

**Figure 1 biology-12-01061-f001:**
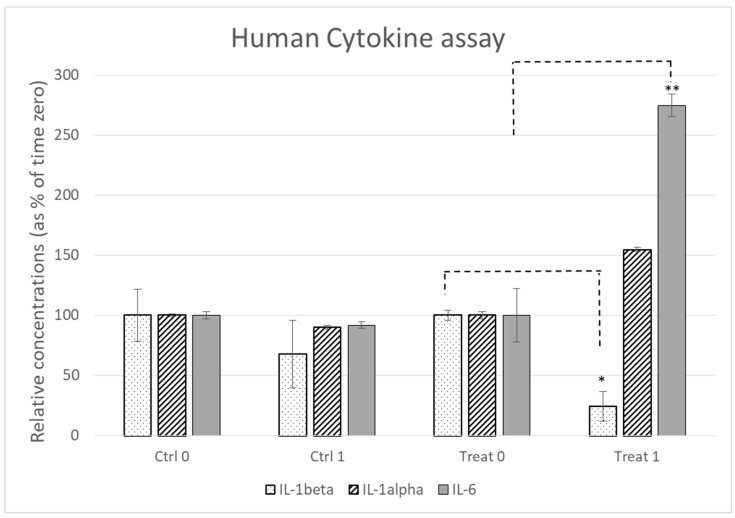
Cytokines’ relative concentration in the two groups at the detection time.

**Figure 2 biology-12-01061-f002:**
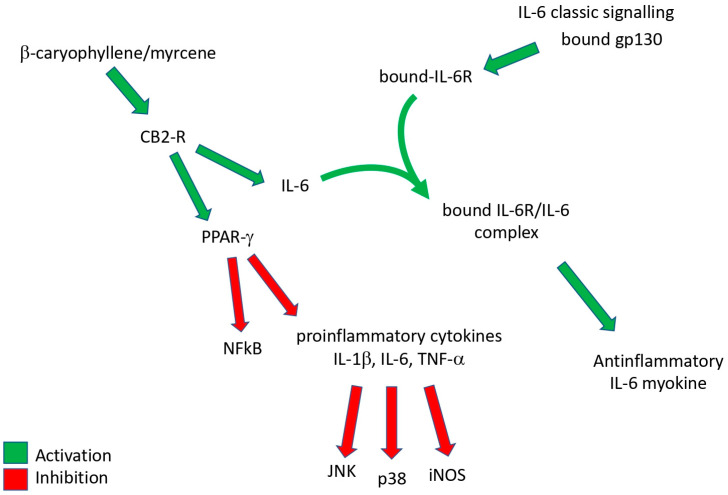
A graphical representation summary of the effects of β-caryophyllene/myrcene on inflammatory cytokines.

**Table 1 biology-12-01061-t001:** Characteristics of the sample, by group.

Variable	Ctrl (n = 19)	Treat (n = 19)	Total (n = 38)	*p*-Value
Females; n (%)	10 (52.6)	10 (52.6)	20 (52.6)	1.000
Age (years); mean ± SD (range)	59.7 ± 6.6 (47–69)	54.5 ± 4.6 (48–65)	57.1 ± 6.2 (47–69)	0.008
BMI; mean ± SD (range)	27.9 ± 3.8 (20.4–33.4)	29.6 ± 6.3 (20.0–49.3)	28.7 ± 5.2 (20.0–49.3)	0.376

Ctrl = control group; treat = treatment group; BMI = body mass index; SD = standard deviation; and n = number.

**Table 2 biology-12-01061-t002:** The absolute values of serum inflammatory cytokines per group at the detection times.

ng/dL	IL-1β	IL-1α	IL-6	IL-2	TNF-α
Ctrl T0	0.0074 ± 0.0016	0.0711 ± 0.001	0.1417 ± 0.0044	n.a.	n.a.
Ctrl T1	0.0050 ± 0.0014 ^NS^	0.0640 ± 0.001 ^NS^	0.1300 ± 0.0038 ^NS^	n.a.	n.a.
Treat T0	0.0650 ± 0.0026	0.0350 ± 0.001	0.0270 ± 0.0060	n.a.	n.a.
Treat T1	0.0159 ± 0.0020 *	0.0541 ± 0.001 ^NS^	0.0742 ± 0.0070 **	n.a.	n.a.

IL-1β = interleukin-1β; IL-1α = interleukin-1α; IL-6 = interleukin-6; IL-2 = interleukin-2; TNF-α = tumor necrosis factor-α; ng/dL = nanograms/deciliters; T0 = time of recruitment; T1 = at the end of the treatment; ctrl = control group, treat = treatment group; NS = not significant; * = *p* < 0.05; ** = *p* < 0.01.

## Data Availability

The datasets used and analyzed during the current study will be made available upon reasonable request to the corresponding author, G.F.

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
