# Peer review of "Effects of Terpenes on the Osteoarthritis Cytokine Profile by Modulation of IL-6: Double Face versus Dark Knight?"

_biology, 2023, doi:10.3390/biology12081061_

Round 1
Reviewer 1 Report
The manuscript titled “Effects of terpenes on osteoarthritis cytokines profile by modulation of IL-6: Double Face versus Dark Knight?” by Giacomo Farì and colleagues aims to evaluate the anti-inflammatory role of two dietary supplements both containing hemp seed oil, but of which only one also contains terpenes in a population of KOA patients. The authors investigated the modulation of serum levels of 5 different pro-inflammatory cytokines (IL-1β, IL-1α, IL-2, IL-6, TNF-α) following 45-days dietary supplement administration and they observed a decrease in IL-1β expression and an increase in IL-6 one. In particular, the increased expression of IL-6 has been associated with the activation of an alternative pathway that has been highlighted only in recent years.
Overall, the manuscript is a well written paper that provides a good background, clearly and extensively discussed results, and critically presented limits to the reader. For this reason, just some minor criticisms need to be addressed to the authors.
In Materials and Methods section (line 169), you specified that the "treated" group took hemp seed oil without cannabinoids. Did the control group take it with cannabinoids? From the text it seems to emerge that the only difference between the two groups is the presence or absence of terpenes. Please clarify in the text specifying the absence of cannabinoids also in the control group.
In Figure 1, it is not clear what the numbers in brackets in the lower part of the figure indicate. Please clarify and expand the figure description in its legend.
I really enjoyed the discussion section. Although it is important for your research to highlight the difference between the systemic chronic inflammation and local low-grade inflammation typical of OA, I find the explanation given in the 354-375 lines to be verbose and rich in unnecessary elements and references. This description makes the reader lose the focus on the result obtained on myokine IL-6. The authors could try to streamline this part by providing instead some extra element on the "double face" of IL-6.
In conclusion section (line 414), did you mean IL-1β?
There are some typos. Minor editing of English language is needed.
Author Response
Point to point letter to reviewer 1.
Dear Reviewer,
We are proud of your positive comments. Thank you so much for them and for Your suggestions, which are great insights for us to improve our work.
Q. In Materials and Methods section (line 169), you specified that the "treated" group took hemp seed oil without cannabinoids. Did the control group take it with cannabinoids? From the text it seems to emerge that the only difference between the two groups is the presence or absence of terpenes. Please clarify in the text specifying the absence of cannabinoids also in the control group.
A. Thank you for this comment. As you correctly understood, the only difference between the two groups is the presence or absence of terpenes, but in the text this was not sufficiently clear, so we better clarified it in the manuscript.
Q. In Figure 1, it is not clear what the numbers in brackets in the lower part of the figure indicate. Please clarify and expand the figure description in its legend.
Q. I really enjoyed the discussion section. Although it is important for your research to highlight the difference between the systemic chronic inflammation and local low-grade inflammation typical of OA, I find the explanation given in the 354-375 lines to be verbose and rich in unnecessary elements and references. This description makes the reader lose the focus on the result obtained on myokine IL-6. The authors could try to streamline this part by providing instead some extra element on the "double face" of IL-6.
A. Thank you for this comment. We have modified the text accordingly, yet we feel that a little bit of contest was necessary.
Q. In conclusion section (line 414), did you mean IL-1β?
A. Thank you for this comment, it was IL-1β, we have edited the manuscript.

Reviewer 2 Report
The present study aims to evaluate the effect of dietary supplements containing hemp seed oil in combination with or without terpenes on cytokine serum levels in patients with osteoarthritis. The Introduction is well-written. It summarized the main goals of the study. The references cited are up to date. The Materials and Methods section contains information about the criteria for selecting included and excluded patients in the study.
The following question arises:
Have the included patients been on any therapy?
Does the dietary supplement of the control group contain cannabinoids and if it is so what is the reason the treatment group does not?
It would be better to describe the procedures of the cytokine assay briefly. What is the min. rate of sensitivity of the used assay?
The authors define the limitations of the presented study by themselves which makes a good impression. The results are a good starting point for future and thorough studies.
Author Response
Point to point letter to reviewer 2
Dear Reviewer,
Thank you for Your comments. We are certain that the quality of our paper will improve following Your precious suggestions.
Q. Have the included patients been on any therapy?
A. Thank you for this question. The included patients had to respect precise criteria about previous treatments, as we reported in the enrollment criteria. As a consequence, any previous therapy was not significant for these subjects at the enrollment.
Q. Does the dietary supplement of the control group contain cannabinoids and if it is so what is the reason the treatment group does not?
A. Both the dietary supplement contains hemp seed oil (not cannabinoids), but only the one for the treatment group contains terpenes also.
Q. It would be better to describe the procedures of the cytokine assay briefly. What is the min. rate of sensitivity of the used assay?
A. Thank you so much for this comment, Bio-Plex Pro Human Cytokine 5-plex assay is designed for the quantification of a panel of 48 cytokines, chemokines, and growth factors in serum, plasma, culture supernatant, and many other sample types. Sensitivity is less than 4 pg/ml.
The authors define the limitations of the presented study by themselves which makes a good impression. The results are a good starting point for future and thorough studies.
A. Thank you so much for this comment, we hope this study could be useful for further studies about dietary supplements.
Best regards